# E-Marketplace as a Tool for the Revitalization of Portuguese Craft Industry: The Design Process in the Development of an Online Platform

**Nuno Martins** [1,*] , **Daniel Brandão** [2] , **Heitor Alvelos** [3] **and Sara Silva** [1]

1   Campus do IPCA, Polytechnic Institute of Cavado and Ave/ID+/FBAUP, School of Design, Vila Frescainha, S. Martinho, 4750-810 Lugar do Aldão, Portugal; fmsarasilva@gmail.com

2   Campus de Gualtar, Institute of Social Sciences, University of Minho/CECS, 4710-057 Braga, Portugal; danielbrandao@ics.uminho.pt

3   Faculty of Fine Arts, University of Porto, Av. de Rodrigues de Freitas 265, 4049-021 Porto, Portugal; halvelos@fba.up.pt

*   Correspondence: nunomartins.com@gmail.com

**Abstract:** The craft trade in Portugal faces challenges that compromise its productive and economic sustainability and may result in the disappearance of millenary techniques, traditions, and industrial practices of high symbolic and historical value. The growing incompatibility of these traditional activities with digital technologies, the lack of resources, and a growing age gap are among the main problems identified. This situation made worse by various restrictions pertaining to the COVID-19 pandemic points towards the possibility of extinction of this type of manual arts. The goal of this research is to demonstrate how the design process of an e-marketplace platform, throughout its different phases, may contribute to the revitalization of traditional industries. The methodologies adopted in the framework consisted in the study of UX and UI best design practices, including wireframe design, user flows, definition of personas, development of prototypes, and style guides. The results of the conducted usability tests to the prototype allowed a gradual improvement of the solution, culminating in the confirmation of its effectiveness. The study concluded that digital technology, namely a designed e-marketplace solution, could potentially bring buyers and sellers closer together, thus being a tool with high potential for the dissemination and sustainability of the craft industry.

**Keywords:** UI/UX Design; Craft Heritage; e-Marketplace; digital design; traditional industries; communication Design; COVID-19 lockdown

## 1. Introduction

This paper presents the different phases of the design process of an efficient and usability tested e-marketplace platform aiming to find out how digital design may contribute to bring crafts producers and consumers closer together.

Traditional industries in Portugal embody its History and Culture, with a particular focus in rural areas. However, these ancient techniques, traditions, and industrial practices are disappearing due to reasons such as a lack of means and resources, an apparent lack of interest from new generations, and poor use of the digital communication and dissemination tools [1]. This niche has made slower progress in embracing the opportunities of the digital world through web-based channels, "compromising livelihoods, devaluing identities and legacies, and nullifying specialised knowledge at times built upon centuries of dedicated practice" [2]. The isolation of these communities has intensified with the situation of the COVID-19 pandemic, since the commerce of handicraft products tends to be

highly dependent on tourism in Portuguese territory. Lockdown, as well as the range of imposed local and global restrictions, have affected tourism and consequently aggravated the precariousness of the handicraft industry, making an adoption of digital and global dissemination tools even more relevant and urgent. Through the creation of an online platform, the present research project aims at connecting craft industry with a global audience, including younger generations, thus contributing to an increase in interest, employment, and improvement of the quality of life of the Portuguese artisans [3].

According to a report by National Craft Registry, there are 4414 registered artisans in Portugal in 2020, 28.6% of which are concentrated Northern Portugal [4]. The current predominant transaction process relies on the physical interaction between the buyer and seller in stores, studios, workshops, and local fairs, as well as word-of-mouth marketing [5]. Through the various testimonies we have obtained through contact with some of these artisans, we have found that they have suffered a progressive downfall in sales of their products, probably due to the current market shifts [1].

The present research therefore arises from the evidence that traditional craft trades in Portugal face difficulties in adapting to new digital technologies. Adapting this traditional craft industry to the advantages of the Internet, particularly network communication and global trade, is proving to be an urgent task [6,7]. However, this adaptation—of vital importance for the sustainability of the craft trade—must be carried out in ways that neither adulterate nor discredit the root identities of Portuguese handcraft production, nor disrupt its workflows. With this in mind, an online platform was designed and will be implemented to allow crafts communities to present and sell their products to a global audience, thus significantly broadening their market base [8].

In this regard, this project seeks to distance itself from other contemporary attempts to help craftmanship by re-shaping their original characteristics with the aim to modernize their systems using new technologies [9]. Its core purpose instead resonates with the UNESCO statement in the "Convention for the Safeguarding of Intangible Cultural Heritage", in 2003:

> "Any efforts to safeguard traditional craftsmanship must not focus on preserving craft objects—no matter how beautiful, precious, rare or important they might be—but on creating conditions that will encourage artisans to continue to produce crafts of all kinds to transmit their skills and knowledge to others [10]."

This project created a website that further displays and disseminates the Portuguese craft practices. It aims to respond to this gap between Portuguese crafts and the current digital setting of global communication, by creating a community-based online platform where artisans can sell their handcrafted products, with a business model initially based on transaction fees. Nevertheless, the digital design process presented in this paper has the potential to be adopted, adjusted, and implemented in other traditional industry contexts, even beyond Portugal.

## 2. Theoretical Framework

### 2.1. The E-Commerce Potential of Craft

As we may observe in Figure 1, electronic commerce has seen a rise of interest in Portugal; however, the traditional craft industries are still quite constrained to an age of globalization [5]. There is an opportunity in digital media to regenerate these practices [2] and help artisans position their small-scale businesses in the global market, as well as optimize their networking [11].

The craftmanship sector in Portugal still relies on "human-centric subsystems" [2], in which selling is taking place at the places of production, craft fairs or, more recently, on electronic commerce [9,12]; exclusive landing pages or presence in e-marketplaces, textile (apparel) and ceramics (decoration) are the categories displaying the most activity [1].

A push towards ethical consumerism with low environmental impact and fair wages also benefits local artisans as their business model effortlessly matches these values. Particularly keen on social issues, younger generations look for sustainable alternatives, and seem willing to pay more for an ethically

sourced product, with a rise of 17% in interest since 2014 [13]. Ethical consumerism thus concerns environmental issues and social justice and is a term that describes a mindset based on moral beliefs that promotes fair trade and alternatives to reduce the negative impact of mass-production industries [14].

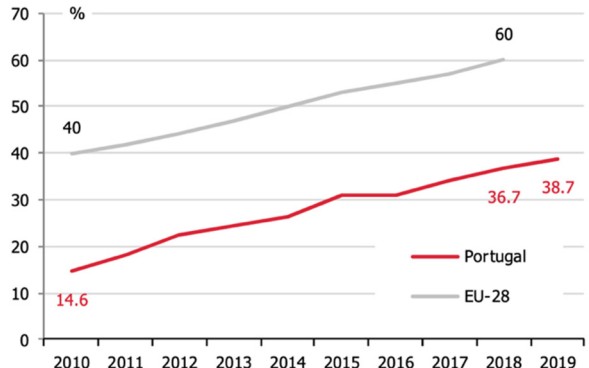

**Figure 1.** Proportion of persons aged 16 to 74 years using e-commerce in 12 months prior to the inquiry, Portugal and EU-28, 2010–2019. National Statistical System (NSS), 2019.

Handmade products are classified as fair trade since they are manually produced with local materials, on a much smaller scale than machine-made products, with adequate price tags [15]. Therefore, this seems fulfils the ideologies of ethical consumerism, endorsing the livelihood of local communities and independent handcrafters for more social equality.

These statements resemble the characterization of the modern craft consumer as being a "selective, conscious of the social, cultural and environmental values associated with handcrafted products and local production, (...) that looks for products with an identity value, being aware of the techniques and quality of the products and contemporary aesthetic" [12].

In the upcoming years, the Professional Training Centre for Crafts and Heritage foresees a significant evolution of this area, affiliating design and new technologies to enhance the potential of handmade manufacturing [16]. Nevertheless, in order to reach out and engage with the target audience, craft practitioners might want to grow an online presence.

According to National Statistical System (NSS), younger generations are the potential target of this project, since the majority (62%) of Portuguese e-buyers are within the age range of twenty-five to thirty-four, and therefore more familiar with new technologies [17]. However, we posit that there are segments of every age group willing to learn how to use new technologies and digital products if it is to their benefit.

Due to the challenge of supporting numerous sellers in building their online presence, there is a need to create a digital platform where every one of them, regardless of their abilities or qualifications, can sign up and sell their products: In this case, an e-marketplace.

### 2.2. E-Marketplace

The e-marketplace is an online infrastructure that allows a community of merchants to conduct commercial transactions [18], such as selling goods or services [19]. This model allows both sellers and buyers to connect in the same digital ecosystem [20] that works as a meeting point or a virtual place [21, 22] and facilitates the communication and efficiency of transactions between both parties [19,23].

These e-marketplaces are often Business to Consumer (B2C) virtual markets, although they also work for Business to Business (B2B) and Consumer to Consumer (C2C) that operate as a neutral intermediary between two parties [22]; the business model is based on charging fees within the platform [24]. These charges, defined by each e-marketplace, can be product listing fees, commissions per sale, processing fees, sign up fees, advertisement (such paid promotions), subscription plans, or events, workshops, and partnerships.

Consumers willingly buy from unknown sellers since they trust the institutional mechanisms of the e-marketplace platform [23]. This concept is far more familiar and less intimidating to the artisan community for its resemblance to traditional physical fairs and dyadic relationships.

Bakos [19] defends that, although hosting multiple participating agents, it is easy to build trusting relationships. Consumers also find this model more beneficial to shop comparatively among multiple sellers without leaving the same website [25], having access to numerous products in one place [24].

Kawa and Wałęsiak [24] listed a series of advantages to the sellers, such as the "marketplace brand recognition" that helps build trust, the "higher number of clients" concentrated in one place, the "additional channel of sales and source of revenue" for handcrafters, "better conditions to promote products" for not being required to invest in Search Engine Optimization (SEO) or advertising, and "access to innovative solutions and analytical and benchmarking data. In this same study, Kawa and Wałęsiak also mention advantages for buyers, such as "higher safety transaction, higher credibility of sellers, possibility to compare prices and offers of various suppliers, feedback on the seller from previous customers" [24]. However, these authors also discuss a series of disadvantages for sellers, such as disproportionate competition from numerous merchants in one place, frequent changes of fees or service costs, and restrictive requirements concerning guidelines with the risk of having their store shut down unexpectedly. Possibly, sellers frequently neglect the option of growing their own e-commerce website on the side, even if performing well within the e-marketplace. From the perspective of a buyer, the main disadvantage is having "shipments from different sellers" that incur in additional fees and different delivery times [24]. Wise and Morrison [26] reinforce these disadvantages because, in this context, the high level of competition might lead to rivalries and lower prices in order to compete against other offers; this might lead to challenges of sustenance.

While an e-marketplace offers products and services from numerous sellers [24,27], e-commerce is a virtual store of a single brand that extends the sales channel of a particular single business. However, the main advantage of an e-marketplace for a seller is that it does not have the costs incurred in conventional e-commerce, such as bureaucracy and large-scale investments related to hosting, domain, inventory management, marketing, search engine optimization, and payment processing.

In e-marketplace, it is essential to build trust and brand loyalty, therefore creating a set of terms and conditions to protect both parties. For example, if consumers have a problem with their order, the seller and the intermediate must take accountability to help solve the issue, with specific terms regarding the return or exchange of products [28]. The e-marketplace needs to have a solid purpose so people will recognize it as the primary source to find a specific product in trusted sellers; otherwise, it is not a sustainable model [29].

### 2.3. User Experience Design (UX) and User Interface Design (UI)

For the development of an intuitive digital product, it is indispensable to understand the concepts of User Experience Design (UX) and User Interface Design (UI). They meet different purposes but contribute together to a workflow based on essential concepts and guidelines to successfully design a digital product [30].

Norman [31] describes UX as the way a user interacts with a product, how they perceive it, and learn to use it. According to ISO 9421-11, usability is defined as an "extent to which a product can be used by specified users to achieve specified goals, with effectiveness, efficiency and satisfaction in a specified context of use" [32].

The role of a designer is to reduce the level of complexity as much as possible, in order to create a user-friendly environment [33,34]; they must also understand the target audience, in order to avoid creating unnecessary difficulties or confusion that may cause a negative perception [35].

The goal of User Interface Design is reducing visual noise and create a smooth navigation experience, causing a positive impact and an ease of use [36]. Therefore, UI can influence consumers in purchasing products and services [37,38], because good looking and well-structured websites are often associated with a high-quality service. As Jennings [39] expounds: "Pleasing visuals are important

because they create first impressions which result in a desire to explore further. Many websites fail to do this and consequently are often cluttered and difficult to understand."

In both UX and UI design, the user is placed at the center, in order to achieve a pleasant experience with the target audience in mind [40]. This design approach is called User-Centered Design (UCD). ISO 13407 describes UCD as an approach that develops "interactive systems focused on creating usable systems" [41]. This empathic approach to UX implies that a designer must listen and observe users [42], in order to develop products that will be used by them. Subsequently, the methodology chosen to develop the e-marketplace has to match these principles.

## 3. Methodology

A Design Thinking strategy was therefore implemented in this project, by focusing the work on real needs, identifying and solving problems directly with users, as recommended by Tim Brown [43]. The Design Thinking is a five-step methodology (six if including the implementation phase) that helps to segment and organize the project from the very beginning, in the research phase, until the ending phase, including iteration loops, which include: Observing and understanding, defining problems, explore ideas, developing prototypes, and testing with representative samples [43].

### 3.1. UX Benchmarking

Before developing a prototype, one must understand the current online scenario that inspires the development of a new e-marketplace. With this in mind, the User Experience Questionnaire (UEQ) UX benchmarking was used in order to compare two references of craft-related Portuguese websites using answers from the representative sample. The two websites selected for our UX benchmarking analysis were Feira de Barcelos and Artemix. For this evaluation, and after performing a set of simple actions, the UEQ method by Schrepp et al. [44] uncovered the strengths and weaknesses of the user experience on both websites.

The UEQ survey presents a list of contrasting attributes, from which the user must spontaneously select the keyword that is closer to their opinion regarding a specific aspect of their experience of the website [44]. These attributes correspond to six major scales [45]: Attractiveness, perspicuity, efficiency, dependability, stimulation, and novelty. A representative sample of this project rated their experience using the aforementioned questionnaire, after browsing both websites and performing a set of basic actions. The findings from this UX Benchmarking inspired the features and improvements implemented in our project.

The Feira de Barcelos website turned out to be less attractive to users. According to users, this website had an outdated visual structure and did not provide a mobile version. The majority of users also considered that the website was slow in loading pages (depending on the internet connection); some pointed out a lack of Secure Sockets Layer (SSL) certificate: This influenced the perception of more experienced users who were aware of these standards. Users also tended to feel overwhelmed when browsing the menu in order to find a product and were unable to work out which artisan made the product, as the website does not provide this information. Overall, the perception of the website was that it is not frequently updated and does not convey enough security for users.

The scenario was slightly more positive regarding the Artemix website, with the exception of perspicuity. The attractiveness and stimulation of the website registered a good first impact with the users. However, users had difficulty browsing the categories, as they do not follow a conventional taxonomy and are not presented as a menu. Users had to scroll along a drop-down selection with numerous tags and felt frustrated in the process as it was the process was quite cluttered. Users also mentioned that their prior initial perception of the design changed as they realized it did not have a mobile version. Furthermore, the footer had items that were not aligned, expressing a desire for a breathable and detail-oriented interface. Artemix does not have an SSL certification either.

Regardless of the above, users pointed out that Artemix website was faster and more pleasing than the Feira de Barcelos website, namely due to the appealing interface. They also mentioned that

it was possible to view information on the artisan who made the product (and visit their profile): This was considered a noteworthy feature.

In short, the Artemix website got significantly better results due to its visual impact. Users apparently found Artemix more attractive than Feira de Barcelos: More than 80% claimed that Artemix had a more distinct design, with visual details resembling a stitch, and a more eye-catching color palette. However, neither website is responsive, nor do they have a clear taxonomy and easy to browse structure. Furthermore, neither includes an in-house Customer Relationship Management (CRM) tool.

From this analysis, it was clear that the first impact of a website influences the perception of usability; therefore, for this project, both UX and UI were carefully developed in order to create a very user-friendly website.

The next step was the design of the online platform, where a low-fidelity prototype and a high-fidelity prototype were developed. There are benefits in designing both models for different phases of the project: In the first stage, the low-fidelity prototype should be developed in order to define the entire layout before moving to a more robust visual design; in the next phase, the high-fidelity prototype means to represent the final version of the product, for testing within a representative sample using the appropriate methods.

### 3.2. Prototype

This topic covers three stages of the Design Thinking methodology: Define, ideate, and prototype. Before developing the visual solution, it was necessary to explore, understand, and define the target audience to create archetypes of users and the structure of the website. These users and structure inspired the creation of user flows, a diagram that showcases the journey each user has to take in the prototype and all the possible options. In order to outline the visual structure of all pages, wireframes were sketched in order to get an overview of the sections that were necessary.

The prototyping phase included a first approach using a low-fidelity prototype in order to create a quick visual representation of the final prototype and validate the different design solutions using an A/B testing. Subsequently, this helped to define the essential components developed in the style guide stage, a topic that covers all the guidelines for the development of the high-fidelity prototype, such as branding, color palette, fonts, grids, and components using the Atomic Design strategy.

Ultimately, a reliable visual representation of the end product was created: In the case of this project, it includes two different high-fidelity interactive prototypes that cover both buyers and sellers' scenarios. These were explored in the ideate and define stage, with an explanation of all features that would be later tested with the representative sample of the project.

### 3.3. Personas

In the present context, the term persona refers to a fictional representation of the target audience of a product that should include details such as behaviors, patterns, goals, and environment [46].

As Alan Cooper refers, "you cannot have purposes without people" [46]. Therefore, it was necessary to implement a goal-oriented scenario (use case) based on two user groups (buyers and sellers) that would allow the definition of all the necessary steps, as well as create realistic scenarios for each prototype.

Inspired by the focus group that would later be used for the usability testing, these personas (Table 1) compiled a set of information that was collected in personal interviews. Both archetypes have female names in line with the most represented segment of the sample, and their ages are the arithmetic average of the users' ages.

These personas were based on actual data from 12 users (six buyers and six sellers) that participated in the second session of usability testing. In this case, we have a 28-year old female buyer with the motivations and the means to purchase handmade products, and a 40-year old female seller,

currently struggling to manage her handmade business, is looking to expand her sales nationwide, and possibly internationally, by using digital means.

**Table 1.** Archetypes. Self-source, 2020.

| Archetype | Buyer | Seller |
|---|---|---|
| Name | Filipa Pereira | Maria Brito |
| Age | 28 years old | 40 years old |
| Occupation | Shop assistant | Full-time artisan |
| Behavioural considerations | Aware of the social, cultural, and environmental issues; | Creates handmade feminine apparel and only sells locally through a social media page; |
| | Enthusiast of conscious consum-erism; | Is aware of trends and creates pieces that are attractive to younger generations; |
| | Wants to support national artisans and purchase fair trade handmade products; | Thought about creating a website but that incurred a high investment and third-party plugins. |
| | Interested in purchasing apparel, furniture, and decorative pieces. | |
| Needs and goals | Being able to find any kind of handmade product within the same platform; | Being able to manage her store in an easy way with more national visibility; |
| | Being able to search by location to support local crafters. | Manage incoming orders with automatic shipping labels ready to print. |
| Frustrations | Isn't able to find all the available offers on the internet for not being aggregated in the same platform, therefore, she usually purchases in foreign e-marketplaces (e.g., Etsy); | Isn't able to participate in all major craft fairs for being distant from her hometown; |
| | | Wants to expand her reach online but doesn't understand social media marketing nor has the time to learn; |
| | Doesn't have the flexibility to visit craft fairs and prefers to purchase online; | Isn't able to manage incoming orders through direct messages. |
| Use Case | Purchase handmade apparel | Setup a female apparel store |

### 3.4. Information Architecture

Information Architecture (IA) is an outline of all pages in a website, in order to provide an overview of the dimension of the prototype, as well as serve as a blueprint for the development stage. In the present case, two hierarchical diagrams were designed to define the necessary pages to create user flows.

In the buyers' case, Figure 2 showcases the pages related to the process of signing up and browsing through the categories (or alternatively, via search bar through keywords or filters). It also includes the pages related to the checkout process and account management, besides institutional and legal pages (e.g., Contacts and Privacy Policy, respectively).

In the sellers' case, Figure 3 showcases the pages related to the process of signing up as a seller. The CRM tool has seven tabs: Dashboard, products, orders, messages, analytics, campaigns, and finances. In case of any issues, the user can access a help center with documentation, tutorials, and a ticketing system, based on Nielsen's recommendations of the 10-usability heuristics [47].

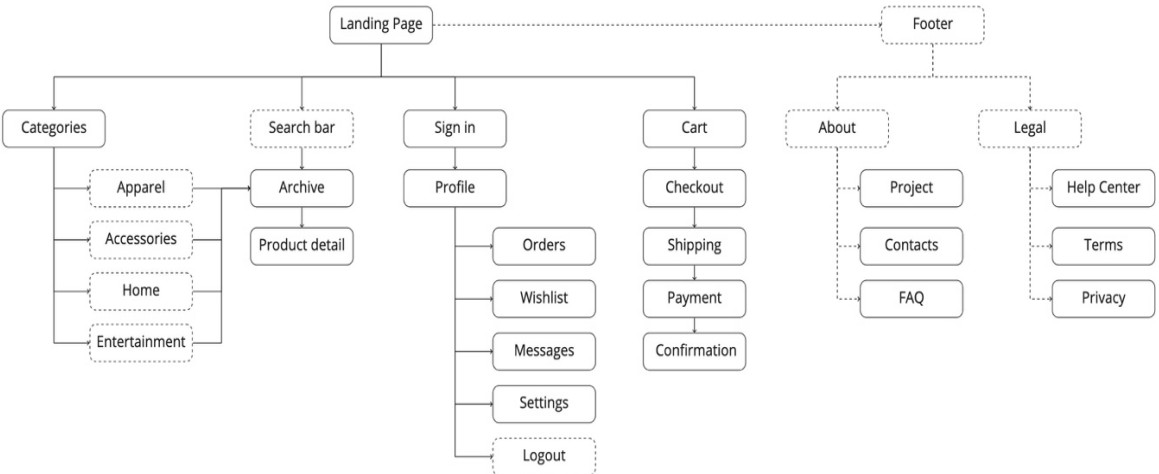

**Figure 2.** Information Architecture (IA) for buyers' prototype. Self-source, 2020.

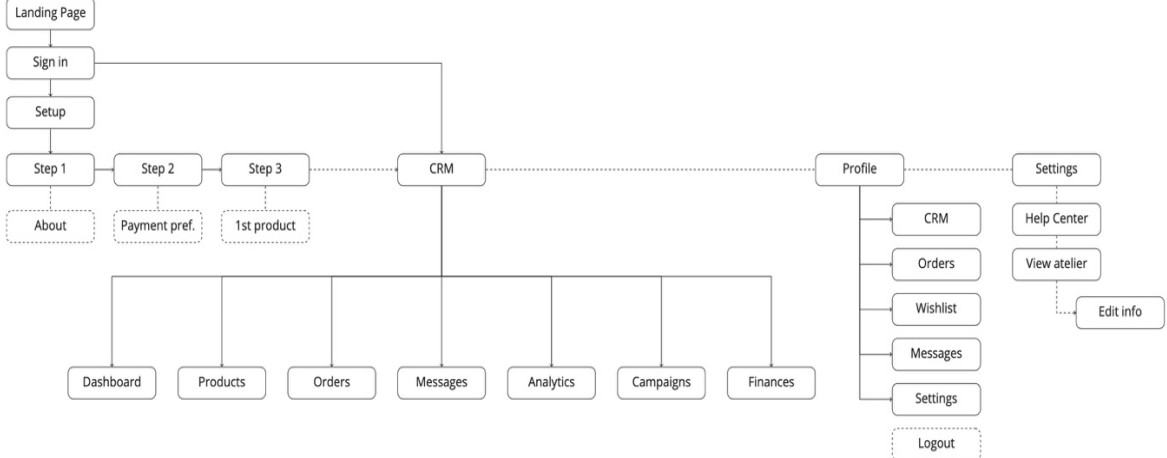

**Figure 3.** IA for sellers' prototype. Self-source, 2020.

### 3.5. User Flows

A user flow is a diagram that showcases all possible pathways a user can take in order to complete a task in a prototype. This phase helps uncover all the necessary steps, from the entry point until reaching the goal established for each case.

The buyers' user flow (Figure 4) showcases a simple routine of purchasing a product on a website. A user can sign in, create an account, or instantly search for a product. Through the search results, the user can filter the products, open a product detail, or visit a store profile, and add the product to a wish list. If the product is in stock, the user can add that item to their cart and checkout.

If the user is already logged in, they can fill the shipping and billing address and payment details (and save them as default for later purchases if they so wish); if not logged in, they can purchase products as guests, or create an account. In the end, the user explores their account to perform the final task and logs out.

With regards to the sellers' prototype (Figure 5), when a user creates a new account, they have to follow a specific three-step flow structure in order to set-up their store. In the first step, the user must fill details about their store, such as name, location, and description; the second step involves configuration of payment preferences for billing. Lastly, they can choose to add their first product, or skip this step and add it at a later stage.

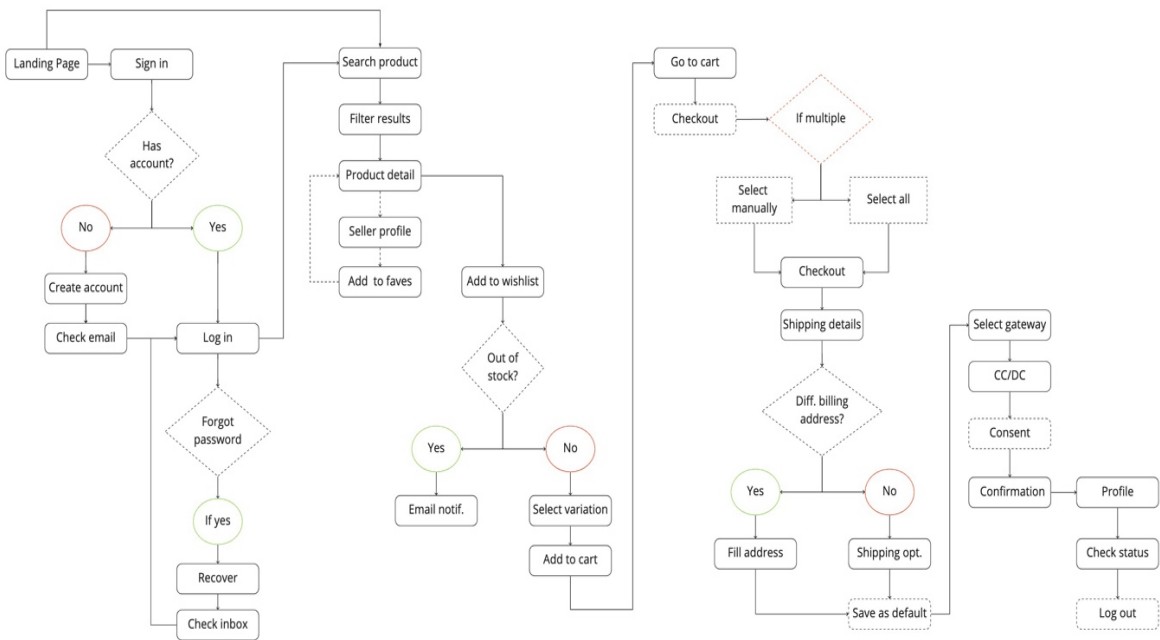

**Figure 4.** User flow for buyers' prototype. Self-source, 2020.

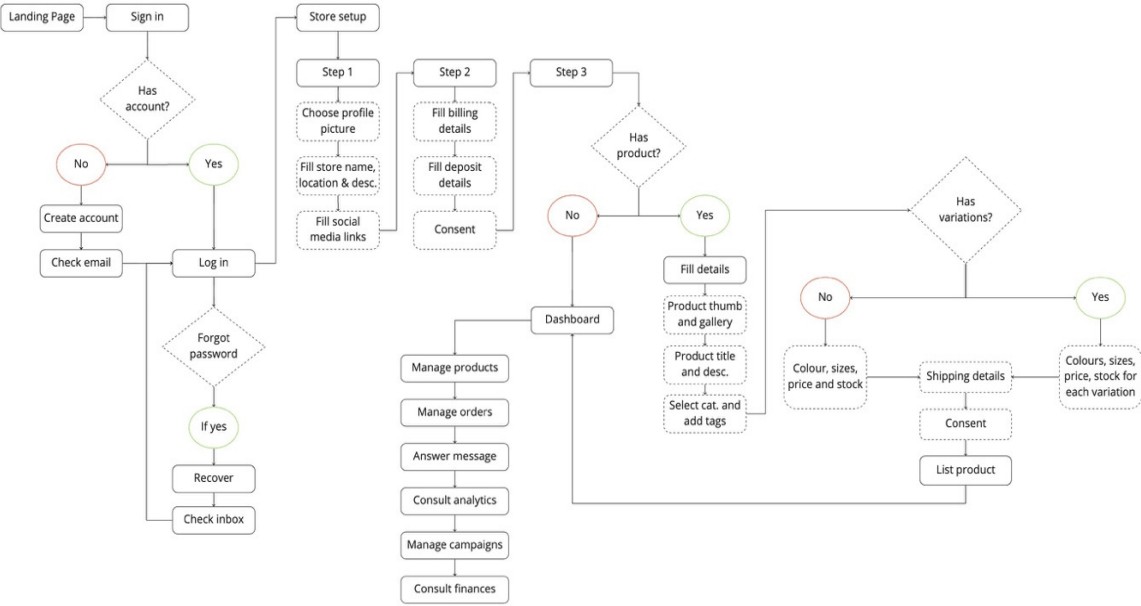

**Figure 5.** Seller user flow. Self-source, 2020.

In the CRM, they can navigate through a set of features, and perform actions such as managing orders and enquiries. It is also possible to view the storefront and edit their own details any time.

*3.6. Wireframes*

As a starting point, wireframes were designed to visually translate the information from the previous steps while bypassing details such as visual design. This method helped quickly iterate and schematize the content hierarchy and interactions between all screens.

For example, at this stage, it was decided that the header should have features such as an unclasped search bar and a full-width menu, a suitable solution to organize all categories and subcategories. The footer would have a call to action before the institutional and legal pages, a list of available payment methods and certificates, plus social media links.

The first sketch was hand-drawn and later interpreted into a more structured, digital version. These wireframes were created with the specific purpose of testing simple user flows and uncovering any issues, before moving to the low-fidelity prototype, where this sketch was used as a structural visual map.

*3.7. Low-Fidelity Prototype*

The visual design process began with a low-fidelity prototype where basic pathways were explored and tested, in order to identify potential usability problems and simplify flows. A small set of minimal blocks were designed, defining the appearance and information each one would contain (for example, product cards). These components were used to build the pages for each flow, readily testing different ways to present the same content and allowing users to decide which solution would be preferable.

3.7.1. A/B Testing

Split testing consists of presenting two versions of the same page, in order to understand which solution works better. This permitted a perception of the efficiency of the design solutions, comparing and validating design solutions and letting users decide, which option they felt more comfortable with. In order to avoid biased opinions in the final usability testing, these tests were conducted with another sample with similar characteristics. The variables namely concern copywriting, placement of elements, colors, and structure.

One of the main issues that were addressed during these tests was the structure of the header, as it contained a considerable amount information to be displayed. In this case, two different approaches were presented for feedback, through an open discussion and observation of the user interacting with the prototype.

As seen in Figure 6, the first header (A) had the categories and search bar separated into different levels within the section, while on the second header (B), the search box was attached to the categories.

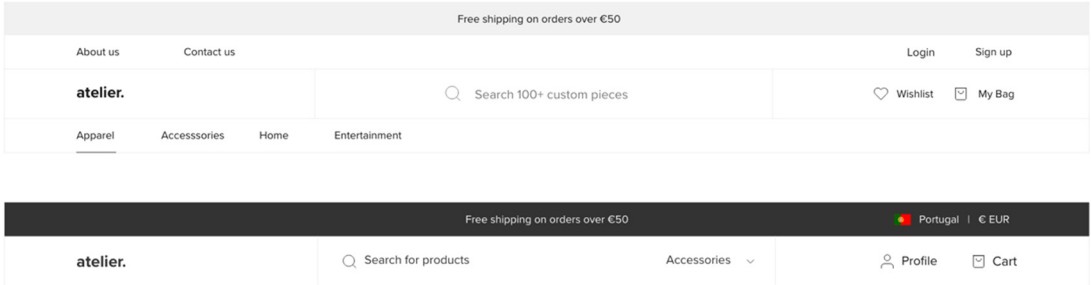

**Figure 6.** First A/B testing. Self-source, 2020.

It became clear during the discussion that the first version (header A) contained too much information and took too much space at the top of the page; users with less experience tended to feel overwhelmed while browsing. Although header B was the most user-friendly option, users questioned whether the top bar was necessary to display warnings, and as such, whether it could be removed.

The second option was therefore considered for a redesign. The currency switcher in the top bar was placed in the footer, and the top bar would only be used to display occasional, relevant messages, such as event promotion, instead of shipping fees, since each seller decides their own fees.

In the second experiment, users felt it would be harder to search with a keyword associated with a category: It would be better to use keywords independently. Additionally, the seller group agreed it would be easier to have a highlighted button in the header for seller sign-up, rather than creating an account and having to switch to a seller profile (this path involving more steps). This feature was implemented, although the seller needs to create a default account to set-up their store and may also switch to a buyer profile.

These tests are a suitable example of how users have a significant role during the design process; with their feedback, any underlying issues were solved before moving to a more accurate visual design, by clearing unnecessary information and sticking to the basics. This minimal approach ended up influencing the overall aesthetic of the prototype.

### 3.7.2. Style Guide

Cooper et al. [33] emphasize the relevance of incorporating design standards, as users appeal to their own memory, by recognizing familiar components in order to easily interact with the website. This improves their understanding, reduces reaction time, and contributes to a positive experience. To accomplish this, the Atomic Design strategy was used to create a cohesive design system with a set of standards employed in the high-fidelity prototype.

Firstly, it was essential to establish a brand identity and color palette that would influence the overall aesthetic of the prototype. For this, a color theory based on the 60/30/10 rule was used for the definition of the primary and secondary colors of the website [48]. The primary color, black, was used across the website to ensure enough contrast with the range of images in prototype; orange was used for the highlights, interactions with buttons (e.g., when hovering a button with the cursor), and illustrations; and light grey was used for small details such as borders.

The font Proxima Nova was used for its suitable proportions and a modern, geometric appearance. Using a base font size of sixteen px (1 rem), the font hierarchy was based on the Golden Ratio for size definition, and line-height for all headings (four out of six were used), paragraphs, captions, and small paragraphs.

To design a layout, it is essential to use a grid to organize all the components on a page. In this case, the grid used was inspired by popular coding frameworks, such as Bootstrap, which uses a default grid of twelve equal width columns with 30-px alleys. This is considered a popular and preferred approach as the number twelve is easily devisable, providing the flexibility to work on different versions of the layout: For example, desktop and mobile responsiveness (stacking website sections vertically), by adjusting how many columns the content takes up [49].

The chosen background color was white, with a grey border to separate sections, enhancing contrast in product images. Likewise, using the 12-column grid, it was possible to provide a column between areas: Hence white spaces were used in order to avoid excessive visual noise.

After defining the design guidelines, the components identified in the low-fidelity stage were designed according to these characteristics, thus ensuring consistency (Figure 7):

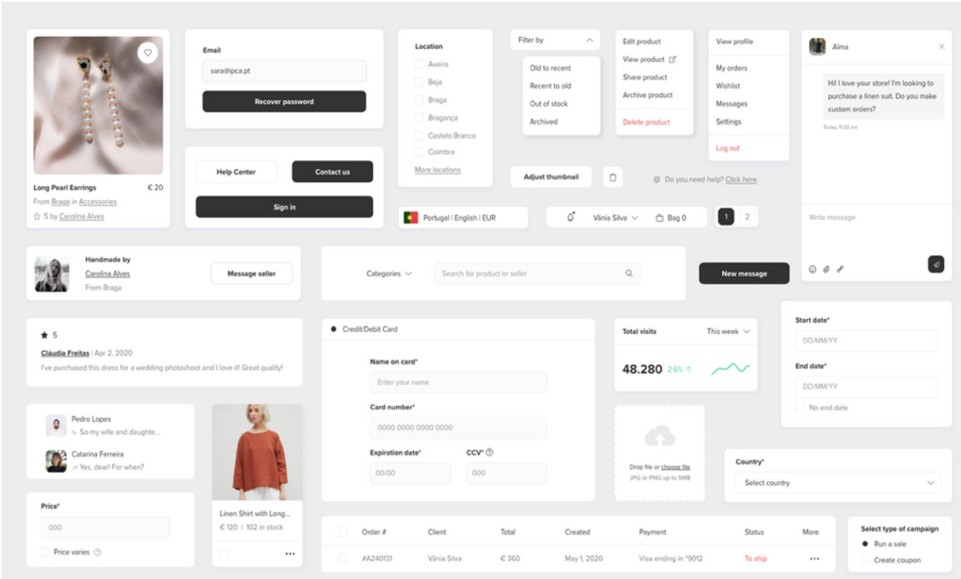

**Figure 7.** A selection of components for the prototype. Self-source, 2020.

In conclusion, Brad Frost's Atomic Design system [50] helped design an original, detail-oriented, and coherent system of nesting components; these were used to build every page in the high-fidelity prototype phase.

*3.8. High-Fidelity Prototype*

The following topics describe the main features that were designed for each prototype and reflect on the aforementioned user flows.

3.8.1. Buyers

The landing page for this prototype was designed according to principles by Steve Krug [51] on creating a high converting website. Krug affirms that a homepage must explain the purpose of the website to newcomers, with a clear overview of its advantages.

Therefore, the homepage (Figure 8) starts with a hero section explaining its mission, followed by a row of recently added products and two sections that respectively explain the advantages and features for sellers and buyers. At the bottom of the page, the footer includes a call to action for the contact and help center, a short description about the e-marketplace, the accepted payment gateways, internal links for secondary pages (e.g., About), and a currency and language switcher.

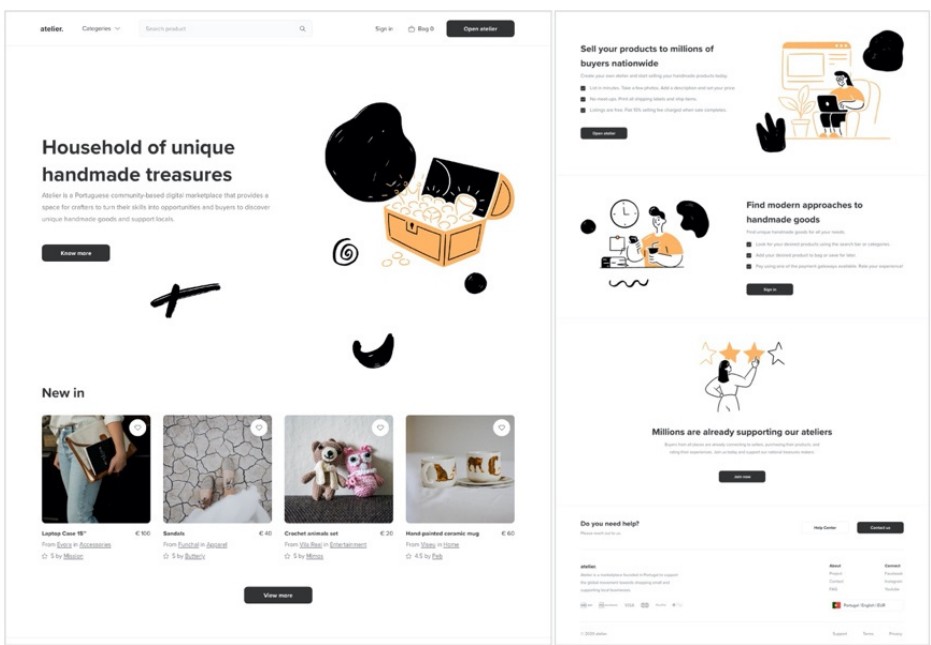

**Figure 8.** Homepage. Self-source, 2020.

It is possible to search for a product without signing in. However, having an account allows the user to access particular features such as order history, wish list, messages, and a push notification system for updates on their favorite products, sellers, and activity.

Buyers can search and filter products using keywords and access more information about a product. In the product detail page, users can check an item's gallery, name, description, price, and stock, along with information about the seller and reviews from previous buyers. At the end of the page, buyers can also check related products (up-selling and cross-selling). If available for purchase, buyers can select changes in the product before adding the it to their cart. If a product is out of stock, the buyer can enter their email in order to receive a notification once and if the product is back in stock.

Consumers can also send direct enquiries to sellers using a pop-up chat available in the product detail page. They may also visit the profile of a seller, including information, a list of products, reviews, Frequently Asked Questions (FAQ), and shipping and return policy.

The checkout includes three steps: An identification form, a shipping and billing address form, and payment options. This process ends with a successful transaction message containing order number and details. If the shipping method provided by the seller includes a tracking number, the buyer will receive that information once their order is shipped.

### 3.8.2. Sellers

Sellers set up their store with information about their business and preferred methods for billing. To list a product for sale, the user must upload photographs of the item and enter its name, description, product category, price, and stock. However, in the case of variations, sellers must specify different combinations of sizes, colors, prices and stock. Each seller is also free to decide the shipping costs for their products.

To manage their store (Figure 9), sellers use a CRM tool with seven operational tabs. The first one, the Dashboard, presents an overview of the store's performance. In the tab Products, users can manage their inventory by adding, editing, or removing products, while the tab Orders enables users to manage incoming orders and print shipping labels.

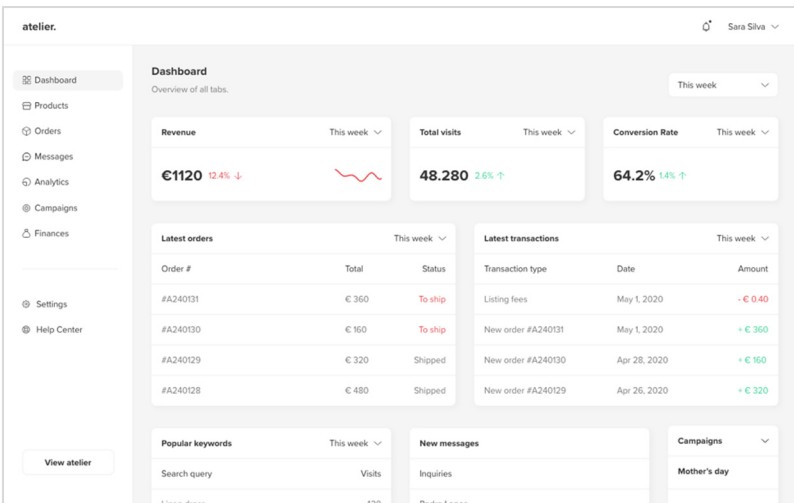

**Figure 9.** Sellers: Dashboard. Self-source, 2020.

To communicate with new leads or customers, sellers can use the tab Messages, containing an optimized messaging system that includes text formatting, emojis, attachments, and links. Sellers can also report or block unsolicited users.

The tab Analytics allows users to access statistical data containing the number of total visits, conversion rates, bounce rates, most popular products and keywords, top locations and demographics, traffic sources, and satisfaction level from reviews. These can be filtered by week, month, current or previous year, or since launch—helping create targeted promotions that can be managed in the Campaigns tab, where sellers can generate discount codes or create a sale with a start and end date. Lastly, the Finances tab presents the store's revenue and expenses; users can change the settings for billing and deposit, as well as download reports for accounting.

### 3.9. Usability Testing

The prototype was prepared for testing with all the flows and interactions required to complete the use cases. Different pathways were created so users would not feel stuck during a task. These scenarios were created so the mediator could observe how the users perform each task in a realistic scenario [52].

At the beginning of the session, the mediator explained the purpose of the project and usability testing, using a friendly approach in order to let the participants feel more comfortable in sharing their opinions. A Typeform link was shared with the sample, and each user had to answer a pre-test

questionnaire that, in the case of buyers, helped evaluate their familiarity with online shopping and interest in purchasing national handmade products; in turn, regarding sellers, the purpose was to understand their levels of digital literacy and their current selling and marketing strategies. This was necessary as the level of digital literacy could easily influence the way users interact with the prototype, as it was perceived during the tests: Users that were not familiar with online shopping were taking longer times to complete certain tasks than a user that was.

Upon fulfilling this questionnaire, each user was provided with access to a link with the prototype, where they were invited to complete the tasks indicated in Typeform. After completing a task, a pop-up message appeared in the prototype, inviting the user to click on "Done", thus moving to the following task. However, just before moving forward, the user had to rate, on a SEQ Likert-scale, the level of difficulty in completing the previous action. The Typeform had logic jumps for cases where the users rated their experience below four. In those situations, the questionnaire would jump to a question where users could comment about their experience.

The mediator never interrupted or intervened in the thought process of the users. In these tests, the Concurrent Think Aloud (CTA) method was employed: Users are encouraged to comment on their experience during the test, so the mediator can register their behaviors. If a user gets stuck in a task, the mediator can solve the conflict by sharing clues or exact steps, depending on the case in question.

In the seller scenario, some tasks did not have many interactive elements on the prototype, such as the pages "Analytics" and "Finances". In these cases, the tasks were replaced with questions, so the user had to consult the answer in the prototype and choose the correct option in Typeform. For example, one of the questions was about the revenue of the previous month; this implied a filtering of data by "previous month" in the prototype in order to get the correct answer.

Afterwards, each user had to rate their experience and opinions of the project and features through a System Usability Scale (SUS) questionnaire. The user could also share further feedback before submitting their form. Before closing the usability session, the mediator took notes throughout an open discussion where users further spoke about their experience and suggested improvements.

## 4. Results

### 4.1. Results of Usability Tests of the First Session

The first usability test was carried out online with a non-representative sample of BA design students from the Polytechnic Institute of Cavado and Ave (IPCA). This session intended to provide a preliminary prototype evaluation, in a beta testing format, and did not represent the entire universe of potential users of the platform. A total of 20 participants, within the age range of 21 to 27, tested both prototypes, from which 85% were female and 15% were male, and provided feedback regarding the experience and the aesthetics of the website. In this context, the purpose of this sample was to evaluate the interest of younger generations in handicrafts, especially in the design field, in an open discussion format.

A unique flow was created so users would experience the entire prototype. Although five users would suffice, Fu, Salvendy, and Turley [53] defend that, in a first phase, it is always desirable to test with as many users as possible in order to uncover around 98% of usability problems and obtain broader feedback.

From the pre-test questionnaire, it was possible to collect data that supported the purpose of the project. More than 60% agreed that purchasing handmade products helps preserve traditional industries, although 33.3% do not know where to find national handcrafts online. About 61.9% were not aware of the existence of websites such as Feira de Barcelos and Artemix; therefore, they would look for handmade goods on Etsy or in social media pages. Further, 90.5% agreed that Portuguese handcrafts have a more authentic appearance than mass-made products.

It was evident that familiarity with online shopping could heavily influence the sample users' experience. For example, a user was not sure of how to add a product to their wish list, as they were not familiar with the heart icon often being used for the purpose.

One of the main issues discussed was the grey color used in borders and boxes. Although some monitors could render the color, most users could not because of their ICC profiles, and thus had trouble navigating through the prototype and performing actions; this generated a level frustration. However, the issue was easily managed through the mediator's explanation that this situation was deliberately caused in order to test different shades of grey. As a consequence, the 4:5:1 rule was implemented "(...) to make sure that users can read text that is presented over a background" and have more contrast between elements [54]. This issue was solved before the second usability testing session, by opting for one of the darker shades of grey that were tested. During the test, when a user identified an issue or technical constraint, they would notify the mediator, who would explain how to proceed without frustration. These tests were therefore useful to solve specific issues before the final session of usability testing.

After completing the post-test SUS evaluation, users were invited to answer questions regarding the purpose of the website and its use. Around 81% would use this website to purchase gifts or products for themselves, out of which 57.1% mentioned that they would be interested in buying handmade products with the sole purpose to support national artisans; 23.8% would be interested in opening their store, while 14.3% would use the website to find artisans for their own projects.

Several design changes were made according to suggestions written by the users; for example, the variations box was not sufficiently apparent, and users suggested that it would be easier if, in case the person enters different sizes and colors in previous steps, the variations option could be selected automatically. These suggestions were useful in improving the UX of the CRM tool.

Overall, the feedback was very positive regarding the purpose of the website, the implemented features, and usability. However, according to both SEQ and SUS results, various conflicting tasks had technical aspects improved for the subsequent tests with the representative sample.

The SUS results ranged from 70 (B) up to 87.5 (A), this value being the highest: The range shows that, even if the user experience was mostly positive, there is always room for improvement. Having a sample of design students with visual sensibility and training contributed to a set of specific improvements. Despite the positive results of the first test, it is worth remembering it did not cover older age groups that tend to have more difficulty in the use of digital media. Therefore, necessary improvements were implemented in order to ensure that the website was as inclusive as possible.

*4.2. Results of Usability Tests of the Second Session*

The second round of usability testing was held online in June 2020, with a representative sample of 12 users of different ages, occupations, and contents; out of these, six users represented the buyer persona and six represented the seller persona, each group testing the corresponding prototype. Tables 2 and 3 present the characteristics of the users who carried out the tests, on the buyers and sellers' sides, respectively.

The session was held through video conferencing in two distinct days, one for buyers and one for sellers. In the case of sellers, users 3 and 6 had the assistance of a younger person but were comfortable enough to share their honest opinions throughout the experience.

Each test took approximately 30 min to be completed, in addition to the pre-test and post-test questionnaires. The data collected through observations, comments, questionnaires, and open discussion provided an understanding of how the suggestions in the first session improved the user experience.

Collaborating closely with users helped identify a set of other possible usability improvements that were not previously considered: For example, design changes were made, according to feedback

and results. After the test, the mediator collected all the answers through Typeform and calculated the SUS score. The results of the second usability test are presented on Tables 4 and 5.

**Table 2.** Buyer sample. Self-source, 2020.

| User | Demographics | Occupation | Current Situation |
|---|---|---|---|
| 1 | Female, 19 years old | Student | Interested in supporting local crafters |
| 2 | Female, 24 years old | Student worker | Purchases in foreigner e-marketplaces |
| 3 | Male, 27 years old | Shop assistant | Purchases in fairs and retailers |
| 4 | Female, 31 years old | Nurse | Purchases through social media |
| 5 | Male, 36 years old | Not disclosed | Purchases in fairs and social media |
| 6 | Female, 45 years old | Seamstress | Purchases in fairs |

**Table 3.** Seller sample. Self-source, 2020.

| User | Demographics | Occupation | Current Situation |
|---|---|---|---|
| 1 | Female, 18 years old | Student/Potential seller | Not selling yet |
| 2 | Female, 23 years old | Helps artisan grandfather | Selling through social media |
| 3 | Male, 34 years old | Handcrafts as hobby | Selling physically and locally |
| 4 | Female, 43 years old | Part-time artisan | Selling through social media and fairs |
| 5 | Female, 57 years old | Part-time artisan | Selling in fairs and locally |
| 6 | Male, 68 years old | Retired/Full-time artisan | Selling in physical store and fairs |

**Table 4.** System Usability Scale (SUS) scores for buyers. Self-source, 2020.

| User | Score | Grade | Adjective | Acceptable | NPS [1] |
|---|---|---|---|---|---|
| 1 | 92.5 | A | Best imaginable | Acceptable | Promoter |
| 2 | 87.5 | A | Best imaginable | Acceptable | Promoter |
| 3 | 77.5 | B | Good | Acceptable | Passive |
| 4 | 82.5 | A | Excellent | Acceptable | Promoter |
| 5 | 92.5 | A | Best imaginable | Acceptable | Promoter |
| 6 | 80 | B | Excellent | Acceptable | Passive |

[1] Net Promoter Score: measures customer experience and predicts business growth.

**Table 5.** SUS scores for sellers. Self-source, 2020.

| User | Score | Grade | Adjective | Acceptable | NPS [1] |
|---|---|---|---|---|---|
| 1 | 95 | A | Best imaginable | Acceptable | Promoter |
| 2 | 92.5 | A | Best imaginable | Acceptable | Promoter |
| 3 | 87.5 | A | Best imaginable | Acceptable | Promoter |
| 4 | 95 | A | Best imaginable | Acceptable | Promoter |
| 5 | 85 | A | Excellent | Acceptable | Promoter |
| 6 | 80 | B | Excellent | Acceptable | Passive |

[1] Net Promoter Score: measures customer experience and predicts business growth.

These SUS scores confirmed that the feedback from the first usability session helped improve the user experience; it proved to be more pleasing with this sample, even considering the different ranges of digital literacy. Both results provide a reliable way to evaluate the usability of the website, alongside heatmaps, with an overall score above 68 (C) in both, which is considered as a positive user experience, with results above 85 (A) being considered the most favorable. Comparing both Tables 4 and 5, it is possible to validate that the prototype for buyers had some conflicting issues that influenced the user perception of the website, with a result as low as 77.5, proving that there was space for improvement, while the prototype for sellers had fewer and minor issues to be addressed.

It was clear that users with a higher experience of the internet did not encounter as many issues as users with less experience. Regardless of the score, the group of sellers were able to navigate through

the prototype with minimal intervention from the mediator, while the group of buyers provided a series of comments, especially users who encountered difficulties in a specific task. An example is the two users who participated with the help of a younger person and that influenced their answer in the SUS question four and seven.

In the Single Ease Question (SEQ), the most conflicting tasks were identified; these results complemented the heatmaps that revealed the areas that retained the users' attention, further showcasing the thinking process of each user in order to complete a task. The collected data consequently inspired various improvements in the UX and UI of the prototype.

### 4.3. Design Changes

This section presents the design changes based on users' written and direct feedback, as well as conclusions from the SUS and SEQ questionnaire. The first design revision concerned the way buyers could receive updates from their favorite stores. The SEQ evaluation had an average of 5.6, a low score when compared with results from other tasks. Furthermore, the heatmap revealed users tended to feel lost in looking for right place to click, thus exposing a UX issue (Figure 10).

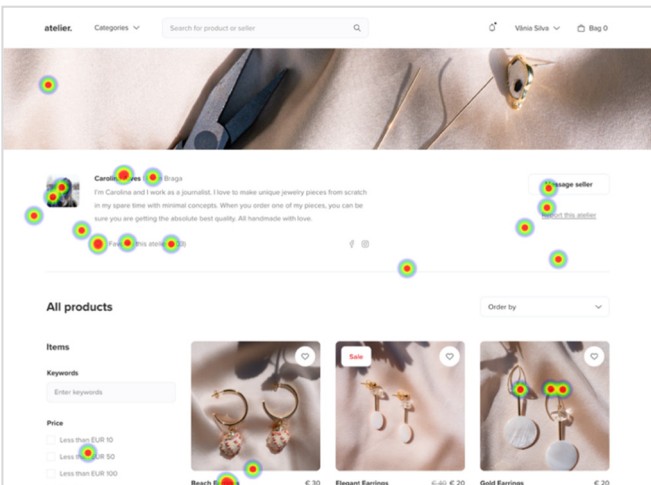

**Figure 10.** Heatmap. Self-source, 2020.

In the original design, below the Biography section, there was a simple text button without borders, presenting a heart icon along with the text "add atelier to favourites" and the number of current followers; this design solution caused a degree of confusion, as it was not easily perceivable that it was a button for being just plain text. Four users had difficulty finding this option.

According to user 2 from the buyer group, the copywriting was not clear enough:

> I had trouble in the task to add her "atelier" to my favourites (…) I'm not sure what it meant (…) Is like following her? (…) it would be better if you can just "follow" her like in social media (…) I think people would like to follow to receive updates (if she has a sale or new products in her store).

As a consequence, this section was tweaked by removing the icon and placing a light, more intuitive button with the word "follow" (Figure 11). Additionally, in order to avoid unnecessary competition and as the purpose of this action is to receive updates about the store, the number of followers could be omitted.

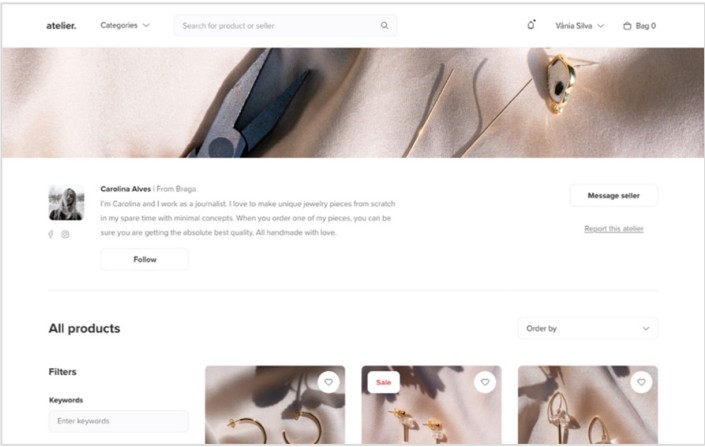

**Figure 11.** Design change: Follow button. Self-source, 2020.

A second suggestion was provided by two users, concerning the way buyers could visit the seller profile. When in a product detail page, right below the attributes of a product, there was a component where users could click to visit the seller profile. However, there was another button close to it that opened a chat pop-up for sending messages to the seller. By having a dark color, it called the attention of these two users that intuitively clicked on it.

The button was therefore replaced with a "visit store" option, while the quick message button, as a secondary feature, was placed below as a single component (Figure 12). A small button with a dialogue icon was considered, but users found that text would be a better option for being more inclusive of a range of users. The buyer can also click on the name or photo to be redirected to the seller's page.

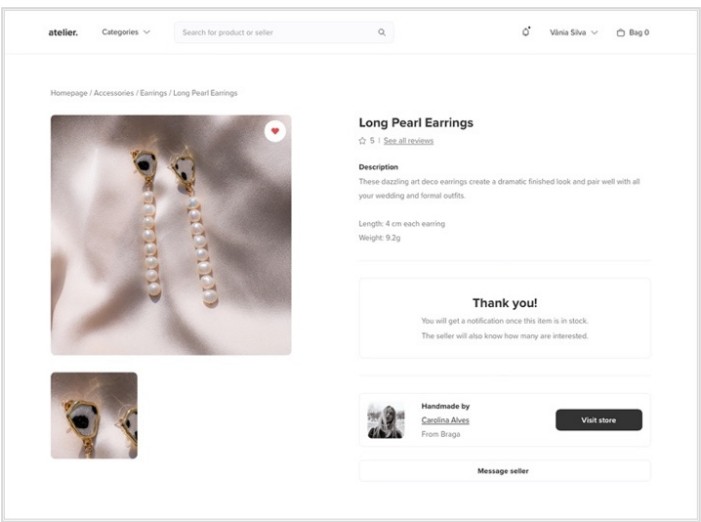

**Figure 12.** Design change: Visit store button. Self-source, 2020.

The last design change in this prototype was the profile view option. The previous design had the name of the buyer (when logged in), along with a down chevron and a bell icon for notifications. Users argued it would be more intuitive to have a picture placeholder similar to "social media" (said user 1), as internet users are generally familiar with social media, and this feature can therefore influence other web experiences. The user can also upload their photo: This feature can be used in the review system, to provide more "credibility for being from a specific person", as user 5 said. Besides this change, the copy of the checkout button was changed to "cart", instead of "bag", as it tends to be a more familiar word in the process of shopping physically.

With regards to the group of sellers, few modifications were made because, as seen in the SUS evaluation, the previously alterations made seemed sufficient to improve the user experience up to the desired standard. However, two users mentioned the button in the landing page for opening a store: The previous design had the words "open atelier", which caused relative confusion with the website name. It was thus replaced by "open store", helping in SEO as a consequence. Users also suggested changing the background color of that button to orange for further visibility.

The mediator also discussed if the group of sellers would find an onboarding helpful: This optional tutorial would explain how each feature works and orient the user through every tab before starting to use the tool. Although these users had a positive experience in learning how to use the CRM, reinforcing that the UX and UI were simple enough for them to understand how to proceed, an onboarding was later implemented for less experienced users (Figure 13). However, in case of any issue, sellers can always resort to the help center.

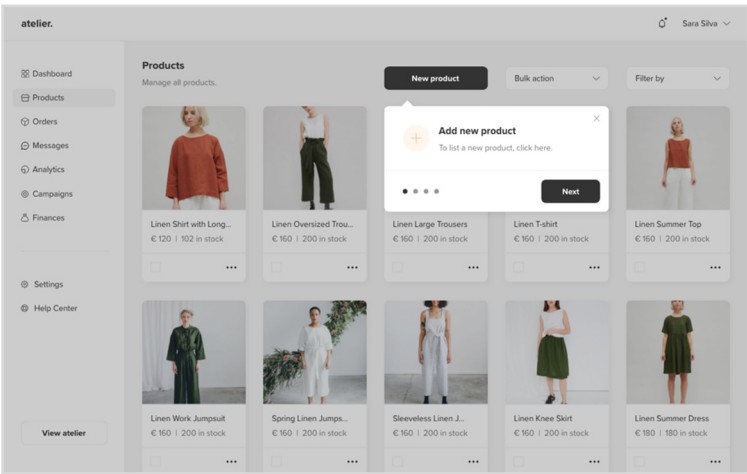

**Figure 13.** Onboarding. Self-source, 2020.

Overall, user experience was very positive. All implemented design changes, except for the first case, were based on user suggestions upon completing the test; these were not necessarily related to usability issues, as every user was able to complete the entire experience. In conclusion, the outcomes of user testing reinforce that the prototypes must be tested with actual users in order to obtain insights onto issues not addressed before due to familiarity in designing the prototype.

## 5. Discussion, Conclusions and Future Work

The purpose of the present research project was to demonstrate how digital design process and its best practices can build efficient online platforms concerning at embracing users with digital literacy constrains and contribute to improve their business activity, by bringing them closer to their target audiences. An e-marketplace prototype was developed with a user-friendly interface and seamless user experience that would encourage Portuguese artisans to sell online effectively. This was achieved through the implementation of user experience and visual design standards in a user-centered approach [55]. The website aims to contribute to the preservation of these practices by providing new income opportunities.

It was clear that this would not be a pioneering solution. However, there is clearly space for improvement within this niche. With this in mind, two benchmarking references were selected, in order to understand the will of artisans to join these platforms, as well as their strengths and weaknesses. We conclude that there is a lack of a dedicated national platform for the management of online craft stores devoid of excessive efforts. During this stage, it was possible to observe how younger generations can lead the way in these traditional craft industries and how recent shifts in consumer behaviors can be a window of opportunity to handcrafts.

Throughout the development of this project, the benefits of direct collaboration with end-users was evident—namely feedback collection and subsequent tool adaptation in accordance with needs and expectations. This approach increases seller's willingness to join e-marketplace strategies, by providing the necessary features in one place, as well as contributing with their suggestions. The contributions of these users were essential in the definition of all development stages of this prototype; hence, the recommendation that the process of decision-making involve the user.

Regarding outcomes, the overall experience was encouraging, proving that the implemented strategies helped create a satisfying user experience: Users did not feel overwhelmed when navigating the final version of the prototype, further expressing an ease of use.

Due to the density of the practical project, a set of experiments and tests have been saved for future iterations. This concerns the web development of a live website, based on the prototype, aiming at launching this digital platform officially in a near future.

After the launch of this platform, the possibility of designing a mobile application is also under consideration: This will help sellers avoid using the responsive mobile version of the website, as it provides limited features on the CRM. For the development of the prototype for this mobile application, industry standards such as Human Interface Guidelines for iOS and Material Design for Android, must be further studied and implemented with a similar flow/scope as this project.

Lastly, a marketing-mix for this e-marketplace is an important path to be explored in future works, in order to position this platform against national competitors, and include new features such as a reciprocal reviewing system where both buyers and sellers can review each other. According to Luca, this helps building trust between both parties, although it can be "potentially biased from fear of retaliation" [56].

The digital design framework presented in this paper can be successfully implemented in other geographical contexts within traditional industry and contribute to overturn the digital literacy barriers and bring consumer and producer of traditional industries closer together. The work presented in this paper is part of a larger project [57] and its impact on the target community in question can only be measured after implementation and a period of effective use, which will be explored in future work. Nevertheless, the main scientific potential of the research presented in this paper is that it proves that a digital design process has to be adjusted regarding each specific context and the aims and constraints of the respective target audience, in order to achieve efficiency and contribute to larger goals, which in this case is the revitalization of industries that are at risk of extinction.

**Author Contributions:** Conceptualization, N.M., S.S. and H.A.; methodology, N.M. and S.S.; web design, S.S.; validation, N.M. and D.B.; formal analysis, S.S.; investigation, N.M., D.B., H.A. and S.S.; resources, S.S.; data curation, S.S.; writing—original draft preparation, N.M. and S.S.; writing—review and editing, N.M., H.A. and D.B.; supervision, N.M.; project administration, H.A.; funding acquisition, H.A. conceptual and linguistic revision, H.A. All authors have read and agreed to the published version of the manuscript.

**Funding:** The Anti-Amnesia Project (POCI-01-0145-ERDF-029022) is co-funded by the Competitiveness and Internationalization Operational Program (POCI), by Portugal 2020 and the European Regional Development Fund (ERDF), and by national funds through FCT—Foundation for Science and Technology.

**Conflicts of Interest:** The authors declare no conflict of interest.

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
