# Peer review of "E-Marketplace as a Tool for the Revitalization of Portuguese Craft Industry: The Design Process in the Development of an Online Platform"

_futureinternet, doi:10.3390/fi12110195_

Round 1

Reviewer 1 Report

Hi Authors, 

Thank you for updating my comments and I am happy to review your article. 

Regards, 

Reviewer 2 Report

The revised manuscript is acceptable for publication in the Journal.

Reviewer 3 Report

I have read the replies of the authors and the revised version of the manuscript, which I believe has adequately addressed the comments of the reviewers.

Also I consulted the editors whether such an applied work is within the journal's scope, and their answer was positive.

So, I gladly recommend accepting the paper and wish the authors all the best in their future work on the project.

This manuscript is a resubmission of an earlier submission. The following is a list of the peer review reports and author responses from that submission.

Round 1

Reviewer 1 Report

The manuscript is interesting and has a potential to be published after major revision:

  1. Abstract and Introduction should contain clear goal of the research. Introduction section should provide more literature review from the period from last three years from the scientific sources like Journals from SCI list, International Journals and Conferences. 
  2. Section 4 should contain sample description (gender and age structure, other relevant information). Analysis of the results is very weak, so I suggest to authors, according to the available data, to provide more scientific results. 
  3. The section 5 should contain scientific contribution of the research.

Reviewer 2 Report

The considered manuscript is dedicated to the development of a website (e-marketplace) for craft items. The paper is well-written and thoroughly follows several established user-centered methodologies. The authors do a good job explaining the project decisions, presenting the designs, etc.

The critical problem that I see with the work is that it's a project report, not a research paper. There are no novel findings, no take-aways besides the actual project artifacts. Meanwhile, Future Internet journal positions itself as "a scholarly open access journal which provides an advanced forum for science and research". I'm afraid that I have to conclude that the submission is outside of the scope of the journal.

Since I might be wrong regarding the unacceptability of project reports to the journal, I am going to leave the special note for the editors. If they do accept non-research papers, this one is good to go with virtually no revisions. The authors would just need to clearly position their work as a project report, and correct the statements regarding the "research", e.g. in p. 19: "The purpose of the present research project was to develop an e-marketplace with a user-friendly interface and seamless user experience"

Reviewer 3 Report

The paper is an interesting approach to the problem of the promotion of products via the internet. Their ideas contribute to the restructuring of the web-based promotion of products in general, although they refer to the Portugese market.

Reviewer 4 Report

Dear Authors, 

Thank you for coming up with this research for Future Internet, I am glad to review and below are my comments. 

  1. According to a report by National Craft Registry, there are 4119 registered artisans in Portugal - Please state the year which this numbers are taken to be transparent
  2. This project created a website that further displays and disseminates the craft practices- give the URL, if you cannot state the reason.
  3. Future research session is missing
  4. Citation are missing in many places
  5. Some of the references are very old, recommend referring 5 years old article unless if it is classic articles related to the current research

Regards,